# High crossreactivity of human T cell responses between Lassa virus lineages

**Brian M. Sullivan**[1‡]*, **Saori Sakabe**[1‡], **Jessica N. Hartnett**[2], **Nhi Ngo**[1], **Augustine Goba**[3,4], **Mambu Momoh**[3,4,5], **John Demby Sandi**[3,4,6], **Lansana Kanneh**[3,4], **Beatrice Cubitt**[1], **Selma D. Garcia**[1], **Brian C. Ware**[1], **Dylan Kotliar**[7], **Refugio Robles-Sikisaka**[1,8], **Karthik Gangavarapu**[9], **Luis Branco**[10], **Philomena Eromon**[11], **Ikponmwosa Odia**[12], **Ephraim Ogbaini-Emovon**[12], **Onikepe Folarin**[11,13], **Sylvanus Okogbenin**[12], **Peter O. Okokhere**[12,14,15], **Christian Happi**[11,12,13], **Juan Carlos de la Torre**[1], **Pardis C. Sabeti**[6], **Kristian G. Andersen**[1,8], **Robert F. Garry**[2], **Donald S. Grant**[3,4,16], **John S. Schieffelin**[17], **Michael B. A. Oldstone**[1]

1 Viral-Immunobiology Laboratory, Department of Immunology and Microbiology, The Scripps Research Institute, La Jolla, California, United States of America, 2 Department of Microbiology and Immunology, Tulane University School of Medicine, New Orleans, Louisiana, United States of America, 3 Viral Hemorrhagic Fever Program, Kenema Government Hospital, Kenema, Sierra Leone, 4 Ministry of Health and Sanitation, Freetown, Sierra Leone, 5 Eastern Polytechnic Institute, Kenema, Sierra Leone, 6 Njala University, Moyamba, Sierra Leone, 7 FAS Center for Systems Biology, Harvard University, The Broad Institute of MIT and Harvard, Cambridge, Massachusetts, United States of America, 8 Scripps Translational Research Institute, The Scripps Research Institute, La Jolla, California, United States of America, 9 Department of Molecular and Experimental Medicine, The Scripps Research Institute, La Jolla, California, United States of America, 10 Zalgen Labs, Germantown, Maryland, United States of America, 11 African Center of Excellence for Genomics of Infectious Disease (ACEGID), Redeemers University, Ede, Nigeria, 12 Institute of Lassa Fever Research and Control, Irrua Specialist Teaching Hospital, Irrua, Nigeria, 13 Department of Biological Sciences, Redeemers University, Ede, Nigeria, 14 Department of Medicine, Irrua Specialist Teaching Hospital, Irrua, Nigeria, 15 Department of Medicine, Faculty of Clinical Sciences, Ambrose Alli University, Ekpoma, Nigeria, 16 College of Medicine and Allied Health Sciences, University of Sierra Leone, Freetown, Sierra Leone, 17 Department of Pediatrics, Tulane University School of Medicine, New Orleans, Louisiana, United States of America

‡ These authors share first authorship of this work.
* bsully@scripps.edu

**Data Availability Statement:** All epitope information (both negative and positive as well as associated HLAs) have been submitted to the

## Abstract

Lassa virus infects hundreds of thousands of people each year across rural West Africa, resulting in a high number of cases of Lassa fever (LF), a febrile disease associated with high morbidity and significant mortality. The lack of approved treatments or interventions underscores the need for an effective vaccine. At least four viral lineages circulate in defined regions throughout West Africa with substantial interlineage nucleotide and amino acid diversity. An effective vaccine should be designed to elicit Lassa virus specific humoral and cell mediated immunity across all lineages. Most current vaccine candidates use only lineage IV antigens encoded by Lassa viruses circulating around Sierra Leone, Liberia, and Guinea but not Nigeria where lineages I-III are found. As previous infection is known to protect against disease from subsequent exposure, we sought to determine whether LF survivors from Nigeria and Sierra Leone harbor memory T cells that respond to lineage IV antigens. Our results indicate a high degree of cross-reactivity of CD8+ T cells from Nigerian LF survivors to lineage IV antigens. In addition, we identified regions within the Lassa virus glycoprotein complex and nucleoprotein that contributed to these responses while T cell

Immune Epitope Database (www.iedb.org) under the following submission IDs: 1000831, 1000832, and 1000836. HLA and some patient information will also be available at the Center for Viral Systems Biology (www.cvisb.org) searchable by patient ID. All flow cytometry data has been uploaded to Flow Repository (http://flowrepository.org/id/FR-FCM-Z2EJ and http://flowrepository.org/id/FR-FCM-Z2E8) and can also be searched using the title of this manuscript.

**Funding:** MBAO received National Institute of Allergy and Infectious Diseases (https://www.niaid.nih.gov) contract grant HHSN272201400048C under BAA-NIAID-DAIT-NIHAI2013167. JSS received National Institute of Allergy and Infectious Diseases grant R01AI123535. The funders had no role in study design, data collection and analysis, decision to publish, or preparation of the manuscript.

**Competing interests:** I have read the journal's policy and the authors of this manuscript have the following competing interests: RFG and LMB are co-founders of Zalgen Labs, LLC. The Viral Hemorrhagic Fever Consortium (vhfc.org) is a partnership of academic and industry scientists who are developing diagnostics, therapeutics and vaccines for LF and other severe diseases. Tulane University and various industry partners have filed United States and foreign patent applications on behalf of the consortium for several of these technologies. If commercial products are developed, consortium members may receive royalties or profits.

epitopes were not widely conserved across our study group. These data are important for current efforts to design effective and efficient vaccine candidates that can elicit protective immunity across all Lassa virus lineages.

## Author summary

Lassa virus (LASV), the causative agent of the hemorrhagic illness Lassa fever (LF), is found throughout West Africa. Humans are usually infected after contact with infected rodent excreta or aerosolized virus. The mortality rate among hospitalized LF cases is high and no effective treatments or vaccines exist. A vaccine effective against the four main lineages of LASV is needed to protect susceptible individuals across West Africa. To understand how this protection could occur, we examined the immune responses of LF survivors from two different regions of West Africa. As previous infection with Lassa virus protects from disease after subsequent exposure, the immune response of LF survivors provides a model of protective immunity that could be induced after vaccination. We found that LASV strains from lineages different from those that infected the LF survivors efficiently activated memory CD8+ T cell responses. We identified regions within LASV proteins that elicit memory responses in the majority of individuals. From these data, we propose that an effective vaccine that protects against lineages across West Africa should be designed to elicit memory CD8+ T cell responses in addition to antibody responses.

## Introduction

Lassa virus (LASV) infects hundreds of thousands of individuals each year in West Africa, resulting in thousands of LF cases with a high case fatality rate among hospitalized individuals with severe LF symptoms. While zoonotic transmission is the main route of human infections [1], nosocomial infections regularly occur[2–4]. The lack of any approved interventions or vaccines make LASV a serious threat to the general public and specifically to health care workers treating Lassa fever patients. There are no FDA-approved LASV vaccines and current anti-LASV therapy is limited to an off-label use of ribavirin that has limited efficacy. LF has been included on the revised list of priority diseases for the WHO R&D Blueprint, and therefore there is an urgent need for accelerated research and development for LASV vaccines[5].

There are at least four distinct LASV lineages circulating in West Africa[1, 6–8]. Though these lineages circulate in geographically distinct regions, an effective vaccine should ideally protect against strains from all LASV lineages. Cross protection across lineages is especially important in Nigeria where three lineages circulate[1, 9]. Studies of cross-reactive adaptive immune responses to LASV are limited. One study identified that several antibodies against the glycoprotein complex neutralized pseudotyped viruses from all four lineages[10]. However, a comprehensive analysis of individual antibody repertoires from LF survivors and how well those antibodies neutralize or protect against reinfection from different lineages has not been done. In addition, some, but not all, LASV-specific CD4+ T cells from lineage IV infected LF survivors responded to antigens from lineage III[11].

Studies evaluating cross protective CD8+ T cell responses to LASV infection are absent. The T cell response during the acute phase of Lassa fever has been associated with both recovery[12, 13] and immunopathology[14–17] and it may be the timing and strength of the T cell response that determines survival.

Immune correlates for protection upon re-exposure or after vaccination may involve both cellular and humoral immunity. LASV-specific monoclonal antibody therapy protects cynomolgus macaques against lethal challenge, even during later stages of disease[18] supporting the idea that boosting humoral responses can contribute to protection against severe outcomes. LASV antigens delivered by a vaccinia vector strongly implicate cell mediated immunity in protection of non-human primates (NHPs)[19] while γ-irradiated LASV did not protect NHPs after challenge despite the generation of LASV-specific antibodies after immunization[20]. Vaccination with an attenuated replication competent vesicular stomatitis virus vector encoding LASV GPC resulted in 100% protection of LASV-infected NHPs and elicited strong cellular and neutralizing antibody responses[21]. Ideally, an effective vaccine should elicit humoral neutralizing antibodies and T cell-mediated protection as both arms of the host immune response are induced in survivors after natural infection.

To better understand the LASV-specific T cell mediated immunity in LF survivors, we generated a library of recombinant single-cycle vesicular stomatitis viruses encoding full and partial regions of LASV glycoprotein complex (GPC) and nucleoprotein (NP) based on lineage IV (Josiah strain). We quantified LASV-specific memory CD4+ and CD8+ T cell responses in 11 Nigerian LF survivors and 37 Sierra Leonean survivors and identified regions in the GPC and NP that elicit broad responses in both survivor populations.

## Results

We generated a collection of recombinant single-cycle infectious vesicular stomatitis viruses (rscVSVs) encoding for LASV GPC and NP antigens (Fig 1A, S1 Table). Genes encoding LASV proteins were inserted into the VSV G locus and viruses were rescued by providing VSV-G *in trans*. We focused on identifying T cell responses against the glycoprotein complex (GPC) and the nucleoprotein (NP) as the vast majority of dominant host responses to mammarenaviruses are directed to these two antigens[22–26]. Replacing VSV G with LASV GPC would yield a recombinant VSV capable of undergoing multiple rounds of infection resulting in the characteristic cytopathic effect (CPE) associated with VSV infection in cultured cells. To minimize the CPE on VSV infected cells presenting antigens (APCs) to T cells[27], we split the gene for LASV GPC into two overlapping segments to cover T cell responses that could span across the S1P cleavage site between $L_{259}$ and $G_{260}$. rscVSV-GP1 encodes for $GPC_{1-279}$, encompassing all of GP1 and the first 20aa of GP2, while rscVSV-GP2 encodes for $GPC_{214-491}$ using the codon for $M_{214}$ naturally present in the Josiah strain of LASV as the start codon.

We confirmed that BHK and peripheral blood mononuclear cells (PBMCs) infected with rscVSVs expressed mRNAs for the genes of interest (Figs 1B & S1A). To measure relative protein expression of antigens of interest in rscVSV infected cells, we incorporated a flag tag at the C-terminus of each gene of interest. We detected protein expression from rscVSV infected cells for all LASV antigens except GP2 (Fig 1C), which likely reflected the high instability of GP2 in the absence of GP1. To overcome this problem, we generated a rscVSV encoding for a stable GP2 to maximize the likelihood that the GP2 protein would be produced in infected cells, taken up by APCs, processed, and presented on MHC Class II to identify LASV-specific CD4+ T cells. For this we fused the stable signal peptide (SSP; $GPC_{1-59}$) to residues 260–491 of GPC ($SSP-GPC_{260-491}$), which resulted in protein expression levels of GP2 readily detected by Western blot (Fig 1C, SSP-GP2).

Individuals with a medical history of previous admission for LF were included in this study. Medical personnel at the Kenema Government Hospital (KGH, Kenema, Sierra Leone) examined all Sierra Leonean subjects and the Human Subjects Committees of the Broad Institute, The Scripps Research Institute, Tulane University's Human Research Protection Program, and

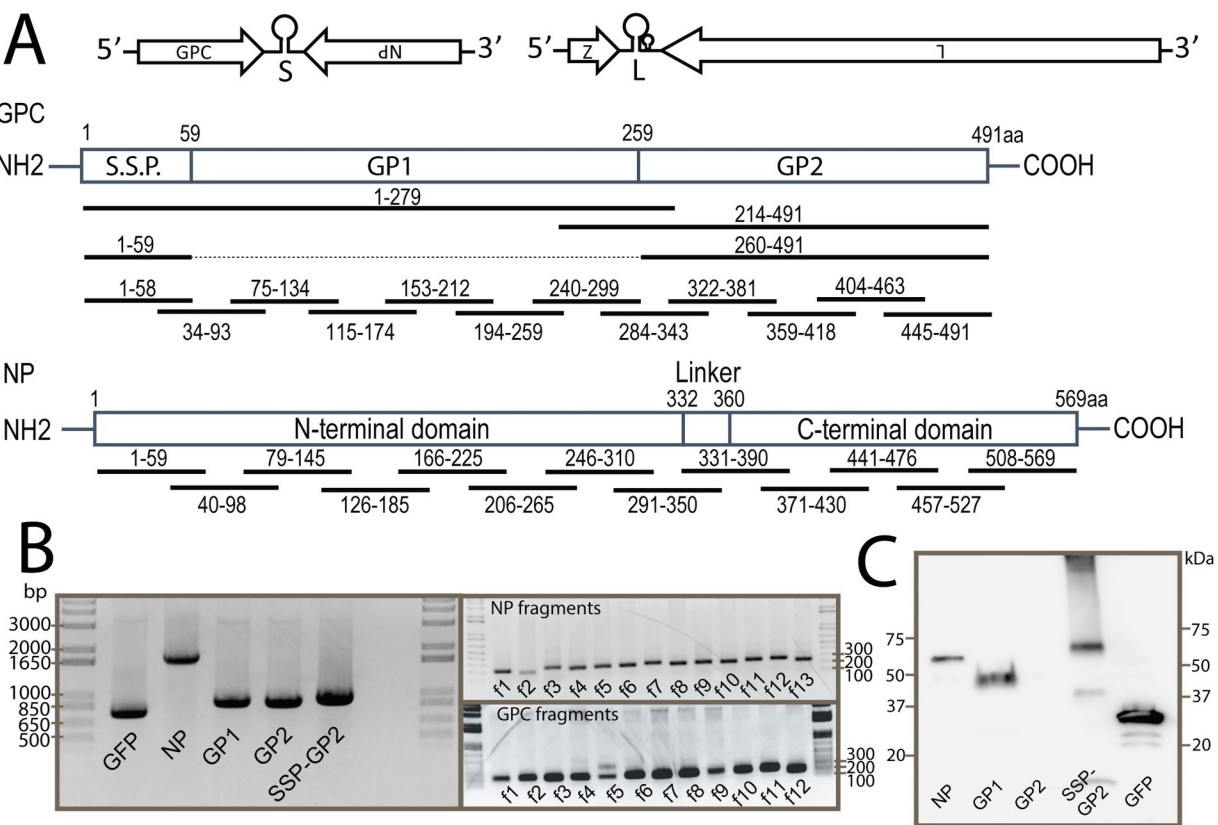

**Fig 1. rscVSVs used to study LASV-specific T cell responses.** A) Schematic of the LASV genome showing the four genes coded in an ambisense direction and non-coding regions (top). Schematic below shows NP and GPC antigens encoded by rscVSVs. B) BHK-21 cells were infected with each rscVSV encoding LASV NP and GPC antigens and GFP control. cDNA was made with poly-dT oligonucleotide primers to amplify only mRNA sequences. LASV gene specific and flag epitope primers were used to amplify cDNA. C) Protein expression of LASV genes and GFP from rscVSV infected BHK-21 cells was determined by Western blot using a flag epitope specific antibody.

the Sierra Leone Ethics and Scientific Review Committee approved this study. Samples from Nigerian LF survivors were collected at the Irrua Specialist Teaching Hospital (ISTH; Irrua, Nigeria) and Nigerian studies were approved by the ISTH Research and Ethics Committee and by the Oyo State Research Ethical Review Committee. Because samples were re-coded at the ISTH, we do not have any information as to which lineage Nigerian survivors were infected. However, genetic surveys done at the ISTH indicate that the majority of patients are infected with lineage II with a minority infected with lineage III viruses[28]. PBMCs were isolated from no more than 30mLs of blood at the Lassa laboratory at the KGH or the ISTH, frozen, and shipped to the United States for T cell analysis.

LASV-specific T cell responses were determined by infecting PBMCs with control or LASV antigen encoding rscVSVs (MOI: 15). At this MOI, the majority of monocytes are infected with minimal numbers of B and T cells infected (S1B Fig) as previously reported[27]. Four hours after infection, brefeldin A (4 μg/mL) was added to cultures followed by overnight incubation. Activation of T cells was assessed through quantification of IFN-γ and TNF-α by flow cytometry (S2 Fig). LASV-specific T cell responses were defined as those samples with a greater percentage of T cells double positive for IFN-γ and TNF-α compared to unstimulated or rscVSV-eGFP stimulated (negative control) PBMCs. Double positive gates were set at 1.2 logs greater than the median of the negative control. For stringency, samples with less than

four IFN-γ and TNF-α double positive cells over the negative controls were considered negative for LASV-specific T cells.

Of the 48 individuals tested, 16 individuals (33.3%) had LASV-specific CD4+ and CD8+ T cell responses, 13 had CD4+ T cell responses and no measurable CD8+ T cell responses, and 19 had CD8+ T cell responses with no measurable CD4+ T cell responses (Figs 2A, S3 & S4). Half of Sierra Leonean survivors (infected with lineage IV viruses), had CD8+ T cells that responded to both NP and GPC constructs while the other half of survivors responded to either NP or GPC (Fig 2B). These results are consistent with observations of similar immunogenicity of NP and GPC of other mammarenavirus infections [29–31], but distinctly different from observations we made of T cell responses of Ebola infections where NP specific responses dominated[32]. CD8+ T cell responses from Nigerian survivors were also equally divided between NP and GPC responses. Surprisingly, we observed that more Nigerian than Sierra Leonean LF survivors harbored CD8+ T cells that responded to both NP and GPC of Josiah strain from LASV lineage IV LASV despite T cells from Nigerian survivors being generated during infection with LASV strains from lineages II and III [1, 33–35]. Of those with CD8+ T cell responses, no significance was found between the percentage of Nigerans and Sierra Leoneans who responded to GPC or to NP (p = 0.22 for both, two-tailed t-test). In contrast, NP dominated the LASV-specific CD4+ response (Fig 2B & 2D; p = 2x10$^{-6}$ comparing combined Sierra Leonean and Nigerian NP to GPC responses, two-tailed t-test).

We quantified the magnitude of CD8+ T cell responses and found no significant differences between antigen specific responses in the Sierra Leonean and Nigerian cohorts (Fig 2C). High basal cytokine expression in some individuals could confound a direct comparison between samples. To minimize this problem, we subtracted the higher cytokine expression levels of the negative controls (unstimulated or rscVSV-EGFP stimulated) from each experimental value. Despite Nigerian survivors having broader responses to LASV antigens, we found no significant differences in the magnitude of these responses between these two groups for each antigen tested (Fig 2E). In contrast, CD4+ T cell responses to NP were significantly higher than responses to GPC (Fig 2D; p = 0.0009 vs GP1, p = 0.0008 vs GP2, mixed-effects analysis).

Thus, we observed similar CD8+ T cell responses to LASV antigens in survivors from Nigeria and Sierra Leone. Next, we asked if the recognized T-cell epitopes were located in common regions within these antigens. To determine regions of high antigenicity, we constructed a library of rscVSVs encoding approximately 60aa polypeptides that overlapped with adjacent regions by 20aa (Fig 1A). When possible, codons for methionine present in the natural antigen were used as start codons (S1 Table). While we could detect mRNA for each of the LASV transcripts in cells infected with each rscVSV (Fig 1B), we could not detect protein expression using an antibody to the flag epitope added to the C-terminus of each LASV gene fragment. However, our previous results using similar methods to identify epitopes from ebolavirus[32], showed that stable protein expression is not necessary for generating peptides for MHC class I presentation.

PBMCs from Sierra Leonean and Nigerian survivors were incubated with rscVSVs encoding for the different GPC and NP 60 aa polypeptides (herein, fragments) in the same manner as with rscVSVs encoding whole antigens described earlier. We found regions in both the NP and the GPC that were over-represented in their ability to elicit CD8+ T cell responses (Fig 3A). These epitope "hotspots" encompassed the carboxy-terminal end of the NP, the carboxy-terminal end of the GPC, and the region encompassing the GP1/GP2 cleavage site. In addition, we found several regions that seemed to be specific to each population. While NP fragment 10 (NP$_{371-430}$) elicited CD8+ T cell responses in two-thirds of Nigerian survivors, we did not detect responses to this region in any Sierra Leonean survivors (p = 0.004, two-tailed t test). In

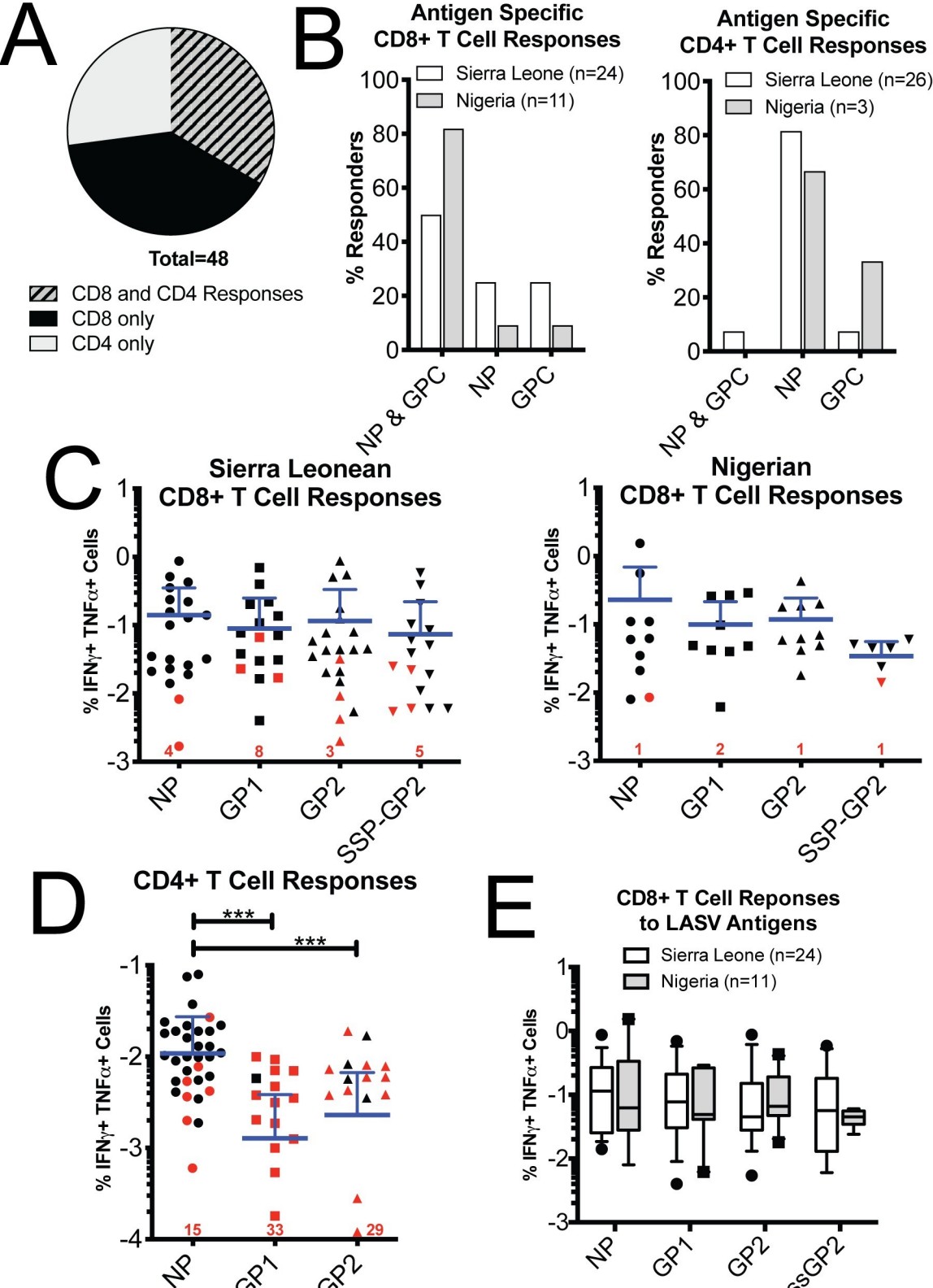

**Fig 2. T cell responses to LASV antigens.** A) Percentages of LF survivors from both Sierra Leone and Nigeria with CD8+ and CD4+ T cell responses to LASV antigens. B) Percentage of LF survivors from Sierra Leone (white bars) and Nigeria (grey bars) harboring CD8 + and CD4+ T cells responding to rscVSVs encoding NP, GPC or both. Responses were defined by individuals expressing both IFN-γ

and TNF-α at 1.2 $\log_{10}$ over the median fluorescence of negative controls. Samples were considered to respond to GPC if we observed T cell expression of IFN-γ and TNF-α upon stimulation with either rscVSVs encoding GP1, GP2, and/or SSP-GP2. Samples were considered to respond to both NP and GPC if we observed T cell expression of IFN-γ and TNF-α to upon stimulation with rscVSV-NP and any of the GPC encoding rscVSVs. Only individuals with CD8+ (left) or CD4+ T cell responses (right) were included. C) Percentages of CD3+ CD8+ T cells expressing IFN-γ and TNF-α from Sierra Leonian (left) and Nigeran (right) LF survivors. Reponses considered positive (black) and negative (red) are shown. Some responses were considered negative because they didn't meet the threshold of >3 events in the double positive quadrant. Zero values are indicated by the numbers above the x-axis. No significance was found between any groups using mixed-effects analysis (Tukey's multiple comparisons test). D) Percentages of CD3+ CD8+ T cells expressing IFN-γ and TNF-α from all LF survivors. Data was analyzed using mixed-effects analysis, **p<0.005. E) Percentages of CD3 + CD8+ T cells expressing IFN-γ and TNF-α from Sierra Leonian Nigeran LF survivors. Only values considered positive for a LASV-specific response are shown. No significance was found between Sierra Leonean and Nigerian responses using 2way ANOVA.

addition, we only detected CD8+ T cell responses to the amino terminal region of the GPC in Sierra Leonean survivors (Fig 3A, GPC fragment 2, p = 0.017, two-tailed t test).

We assessed the magnitude of the CD8+ T cell responses by determining the percentage of CD8+ T cells expressing both IFN-γ and TNF-α in response to each antigen (Fig 3B). The strongest responses also correlated to those areas where responses were observed in most individuals, namely responses to rscVSVs encoding NP fragments 1, 10, 11, and 12 (p = 0.014, 0.046, 0.0005, and 0.034 respectively; one way ANOVA compared to negative control) and GPC fragments 6, 7, and 11 (p = 0.003, 0.026, and <0.0001 respectively; one way ANOVA compared to negative control). The most robust responses to NP fragments (>0.5% of CD8+ T cells) were from a single Nigerian survivor who also had very strong responses to rscVSV encoding NP (Fig 3B, left graph, fragments 7, 12, and 13). In addition, Nigerian survivors also had stronger responses to GPC fragments 6 and 7. Despite these differences, the overall magnitude of the CD8+ T cell response to LASV antigens did not differ significantly between Sierra Leonean and Nigerian survivors (Fig 2E). It should be noted that since we have no information as to when those in the Nigerian cohort had acute disease, we do not know whether lower Sierra Leonean responses are due to a decrease in immunity over time or whether Nigerians mount stronger T cell responses to LASV antigens. In either case, our data indicates that Nigerian responses to lineage IV antigens are at least on par with responses from Sierra Leonean survivors.

Because we designed the fragments encoded by rscVSV to overlap by 20aa, we were able to narrow down the regions in which epitopes occur. We narrowed epitope-containing regions to 20aa if two adjoining regions showed similar CD8+ T cell responses. If a fragment encoding rscVSV inducing CD8+ T cell responses is flanked on each side by fragments that do not elicit responses, we registered the epitope-containing region as the non-overlapping region plus seven amino acids on the amino- and carboxy-terminal sides into the overlapping regions of neighboring fragments. When CD8+ T cell responses were found in response to three adjacent fragments, we did not attempt to define an epitope containing region. We compared deduced epitopes from Nigerian and Sierra Leone survivors and found that while epitopes were concentrated in the N-terminus of the NP, these epitopes were likely distinct (Fig 3C, left panel). We did, however, find more commonality between deduced epitopes in the GPC with the majority of those in the carboxy terminal regions of both the GP1 ($GPC_{240-259}$) and the GP2 ($GPC_{412-451}$) (Fig 3C, right panel).

We sought to validate deduced epitopes from two LF survivors by identifying smaller 10aa epitopes within these larger regions. Fig 4A and 4B show the CD8+ T cell responses to rscVSVs encoding for LASV NP and GPC respectively from Nigerian survivor, N-14, and Sierra Leonean survivor, 5513520 from which deduced epitopes were identified. We used sequences from the deduced epitopes $NP_{139-172}$, derived from a positive response to fragment 4, and $GPC_{412-451}$, derived from a positive response to fragment 11, to identify a set of putative 10aa peptide

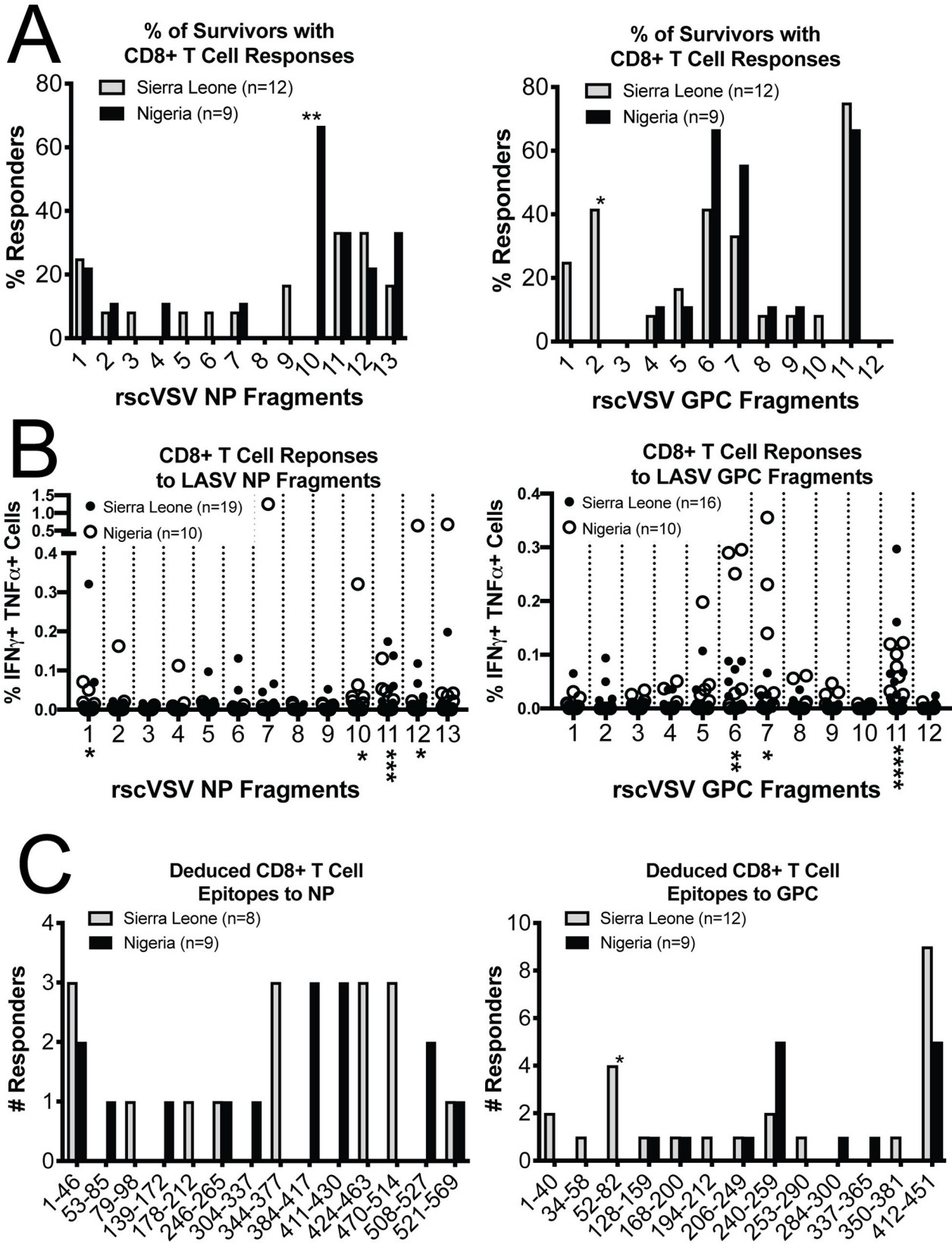

**Fig 3. CD8+ T cell responses to discrete regions within LASV NP and GPC.** A) rscVSVs encoding for ~60aa polypeptides (fragments) from LASV NP and GPC (Josiah strain) were incubated overnight with PBMCs from LF survivors from Sierra Leone (grey) and Nigeria (black) in the presence of brefeldin A. Percentage of individuals who harbor CD3+CD8+ T cells expressing IFN-γ and TNF-α in response to each fragment is shown. Only individuals who responded to at least one LASV antigen are shown. Statistical significance was calculated comparing responses from Sierra Leonean (grey bars) and Nigerian (black bars) survivors using two-tailed t test. (B) Percentages of CD3+ CD8+ T cells expressing IFN-γ and TNF-α subtracted from negative controls from all LF survivors are shown whether or not each individual was considered to have responded to the whole antigen. Responses from Nigerian survivors (open circles) and Sierra Leonean survivors (closed circles) are shown for each rscVSV encoding NP and GPC ~60 amino acid fragments. Statistical significance was calculated using one-way ANOVA (Friedman test). C) Data from CD8+ T cell responses to fragments was used to deduce epitopes. If two adjacent fragments elicited a similar response, the overlapping area was considered an epitope region. If a fragment elicited a response while adjacent fragments elicited a null response, the epitope region was considered to be the non-overlapping regions plus seven amino acids into overlapping regions on either side. Statistics using two-tailed t test comparing Sierra Leonean and Nigerian groups are shown. For all statistical analyses, lack of asterisk indicates no significance; ***$p < 0.0001$, **$p < 0.005$, *$p < 0.05$.

epitopes predicted to bind the HLAs present in the survivors (S2 Table). Using PBMCs from survivors N-14 and 5513520, we found that a $NP_{155-164}$ and $GPC_{440-449}$ elicited CD8+ T cell responses comparable to those from rscVSV whole antigen and fragment stimulations (Fig 4D & 4E).

To examine whether epitope "hotspot" regions that elicited responses from both Sierra Leoneans and Nigerian survivors were more conserved than regions outside of these hotspots, we used amino acid sequences from 600 full-length S segment sequences (Lineage II: 420; lineage III: 49; lineage IV: 131) with an overall amino acid conservation rate of 94.79%. We analyzed four regions that elicited CD8+ T cell responses from multiple Sierra Leonean and Nigerian survivors: two within the NP ($NP_{1-46}$ and $NP_{411-476}$) and two within the GPC ($GPC_{240-259}$ and $GPC_{412-451}$). Overall, we observed a significant difference (p = 0.0313; two-tailed Mann-Whitney test) between amino acid conservation within these regions (96.05%) compared to amino acid conservation outside these regions (94.55%). When analyzed individually, we found differences in conservation in $NP_{1-46}$ (97.6%), $NP_{411-476}$ (94.26%), $GPC_{240-259}$ (99.32%), and $GPC_{412-451}$ (95.23%) compared to conservation of amino acid sequences outside these regions, though none reached significance. $GPC_{440-449}$ contains a similar HLA-A*02-restricted epitope that had been previously identified through animal experiments, although LF survivor 5513520 does not express the HLA*A-02 allele (S2 Table). Though we found little difference in amino acid conservation in the larger deduced epitope region (95.23%) compared to conservation outside epitope hotspot regions (94.55%), amino acids within $GPC_{440-449}$ are nearly universally conserved (99–100%) except for position $GPC_{449}$ where a lysine is conserved only among 67% of sequences (average across $GPC_{440-449}$: 96.11%).

## Discussion

Our results indicate that LASV-specific CD8+ T cells from Nigerian LF survivors recognized antigens from lineage IV (Josiah strain) LASV. To our knowledge this is the first report detailing cross-reactive inter-lineage T cell responses to LASV in humans. Significant differences in LASV sequences have been observed both within [28, 36, 37] and between [1, 7] lineages with up to 32% variation of nucleotide sequences. Despite these differences, we found robust recognition of lineage IV antigens by LASV-specific T cells selected during primary infections with Lineage II and III LASV. While T cell epitopes were found throughout NP and GPC coding regions, several regions were overrepresented in responses from both Nigerian and Sierra Leonean LF survivors; specifically the C-terminal 158 amino acids of NP (represented by NP fragments 11–13), a C-terminal region of the GPC ($GPC_{404-463}$, represented by GPC fragment 11), and the region spanning the GP1/GP2 cleavage site (represented by GPC fragments 6 and 7).

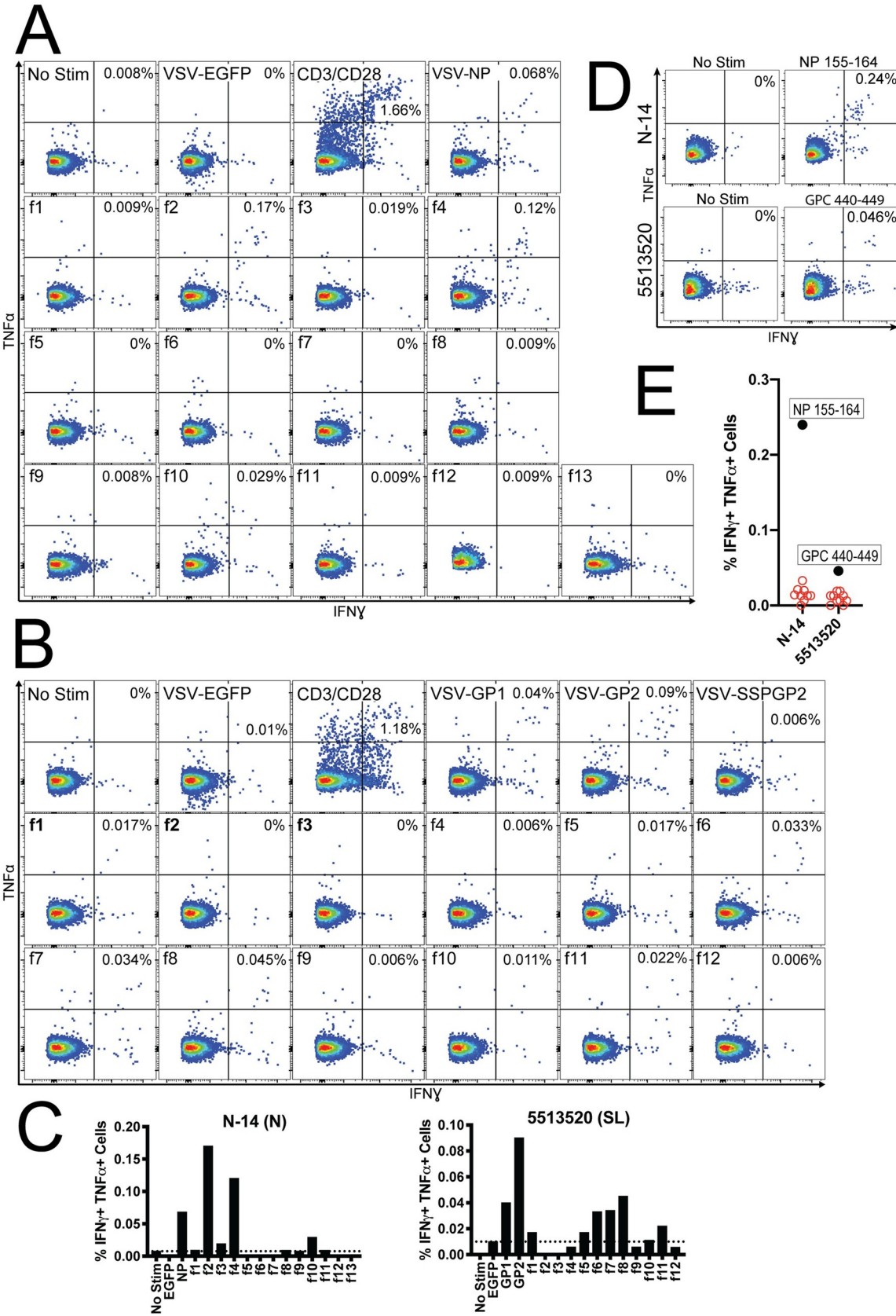

**Fig 4. Identifying 10aa peptide epitopes within deduced epitope regions.** A) PBMCs from a Nigerian LF survivor, N-14, were stimulated with rscVSVs encoding for LASV NP, 60aa fragments derived from NP, anti-CD3/CD28 positive control, and negative controls. Positive responses were observed with stimulations of rscVSVs encoding NP f2, f4, and f10. B) PBMCs from a Sierra Leonean LF survivor, N-14, were stimulated with rscVSVs encoding for LASV GP1, GP2, ssGP2, 60aa fragments derived from GPC, anti-CD3/CD28 positive control, and negative controls. Positive responses were observed with stimulations of rscVSVs encoding GPC f6, f7, f8, and f11. C) Data from flow cytometry plots in A & B are graphed with dotted horizontal lines indicating the threshold for negative responses. D) Flow cytometry plots of CD8+ T cells from N-14 and 5513520 showing positive responses after incubation with either NP$_{155-164}$ and GPC$_{440-449}$ compared to unstimulated controls. E) Data from D (black filled circles) graphed along with other peptides from the same experiment that did not produce CD8+ T cell responses (red open circles).

LASV-specific CD4+ T cell responses were overwhelmingly to NP. As peptides loaded onto MHC class II are processed from exogenous antigens, it is possible that expression differences in rscVSV infected cells skewed the CD4+ responses. We did not, however, observe substantial differences in LASV NP, GP1, and ssp-GP2 expression levels in cell lines infected with the corresponding rscVSV constructs (Fig 1). Overnight stimulation with rscVSV encoded antigens was designed to detect a robust T cell response and may not be sufficiently sensitive to identify low frequency LASV-specific memory T cells in all individuals. In a case study of a Lassa infected individual, memory CD4+ T cell responses were weaker than CD8+ T cell responses four months into convalescence[13]. Lastly, another study measuring CD4+ T cell responses in LF survivors used a more sensitive proliferation assay and found that seven of 13 individuals tested mounted responses to both GP2 and NP[11, 38]. A proliferation and restimulation assay could be used to detect additional virus-specific T cell responses that would not otherwise be observed using overnight stimulation and intracellular cytokine staining.

While peptide matrices are an increasingly common method of identifying specific T cell responses, we chose to deliver antigens through a viral vector for several important reasons. First, we were concerned with the possibility of identifying cross-reactive responses that may not have developed during the immune response to LASV. Populations from developing countries have been shown to have specific differences in T cell differentiation and function compared to those in developed countries, likely due to the high antigenic burden in these regions [39]. A high antigenic burden could generate a more diverse memory T cell repertoire, increasing the likelihood of false-positive, cross-reactive responses. Second, by using peptides alone, we run the risk of non-LASV specific TCRs recognizing non-cognate LASV based peptides as TCRs have shown to have some flexibility in their ability to recognize similar peptides[40–42]. Delivering antigen through rscVSVs retains the biological processes of peptide processing and presentation essential to the development of immune repsonses with the benefit of eliminating potential cross-reactive peptides that would never have been presented in vivo. Third, because of limitations in the amount of each sample available to us, performing T cell stimulations using peptide matrices and subsequent deconvolution and validation would not be feasible.

We defined positive T cell responses as T cells expressing both IFN-γ and TNF-α in response to antigen over negative controls. Single expression of IFN-γ and TNF-α were high in many negative controls and would not provide a robust enough criterion for positive responses. While we measured IL-2, IL-2 expression was often low and the number of cells expressing all three cytokines, though often consistent with samples considered positive, fell below our threshold for the number of cells within the positive gates. Non-normalized data for all CD4+ and CD8+ T cell assays stimulated with rscVSVs encoding whole antigens was included so the reader can assess total cytokine output for each stimulation (S3 and S4 Figs).

We deduced epitopes from T cell responses to rscVSVs encoding for ~60aa fragments of LASV NP and GPC and found that most minimal epitopes in these regions were likely different between Nigerian and Sierra Leonean survivors. Differences in minimal epitopes are not surprising as HLA expression in our cohort between these two populations is diverse,

especially at the HLA-B and -C loci (S5 Fig). Twelve individuals had CD8+ T cells that responded to either GPC fragments 6 and/or 7 that span the GP1/GP2 cleavage site. Deduced epitope analysis revealed that nine of these individuals had responses specific to the carboxy-terminal end of the GP1, while only two individuals responded to the amino-terminus of the GP2. In general, we found that CD8+ T cell responses to carboxy-terminal regions of NP, GP1 and GP2 composed a substantial portion of these responses. We acknowledge that larger deduced epitope regions were not experimentally validated and that epitopes could potentially like outside these regions (but within the ~60aa fragments that were experimentally tested). However, as shown by the identification of two 10aa epitopes in this manuscript and those found for Zaire ebolavirus using similar methods[32], deduced epitope sequences can be used as the basis for identifying smaller peptide epitopes.

We identified four epitope hotspots within LASV NP and GPC. Amino acids within epitope hotspot regions were more widely conserved than amino acids outside these regions with the exception of $NP_{411-476}$. This result was somewhat surprising as these regions are relatively large compared to the size of minimal peptide epitopes. Furthermore, typically only a few amino acids need be conserved to ensure MHC binding and peptide-MHC recognition by TCRs. Like the increase in amino acid conservation we observed from a larger deduced region to a 10aa peptide ($GPC_{412-451}$ vs $GPC_{440-449}$: 95.23% and 96.11%, respectively), we would expect that minimal epitopes in larger deduced epitope regions shared between individuals in different regions would have higher rates of amino acid conservation.

The high genetic diversity and the high incidence of LASV in Nigeria should be considered when designing vaccines and testing their efficacy in populations across West Africa. Most vaccine candidates utilize the lineage IV Josiah strain as the immunogen, and therefore might be able to elicit protective adaptive responses to LASV strains from different lineages. Cross-reactive LASV-specific responses should be assessed, as well as T cell responses followed after immunization with candidate vaccines. Our data suggest that LASV-CD8+ T cell responses can respond to antigens from other lineages to a high degree. Many of these epitopes may be within the antigenic hotspots we identified in the carboxy terminal regions of NP, GP1 and GP2. Lastly, unlike poor CD8+ T cell responses to Zaire ebolavirus GP[32], CD8+ T cell responses to LASV GPC are elicited in a large proportion of LF survivors and these responses are of similar magnitude to those against NP. In summary, we have surveyed dominant T cell responses in the largest cohort of survivors to date, and our data provide the first evidence of broad cross-reactive CD8+ T cell responses in LF survivors. Deploying a single LASV vaccine that protects against infection from all West African lineages could save thousands from developing Lassa fever. Our data indicate that protection across lineages could be enhanced by inducing cell mediated immunity in vaccinated individuals.

## Materials & methods

### Ethics statement

This study was approved by the Human Subjects Committees of the Broad Institute, The Scripps Research Institute, Tulane University's Human Research Protection Program, the Sierra Leone Ethics and Scientific Review Committee, ISTH Research and Ethics Committee, and the Oyo State Research Ethical Review Committee. All adult subjects provided written consent and children participating in these studies were required to have written consent from a parent or guardian.

### Subjects

All subjects have a documented clinical history of Lassa fever. 36 of 48 individuals in this study were also assessed for Lassa virus-specific antibodies (S6 Fig). Human serum reactivity against

Lassa virus glycoprotein was measured with the ReLASV Pan-Lassa Prefusion GP IgG/IgM ELISA Kit ELISA (Zalgen Labs, Cat No. 10580). The kit measures semi-quantitative detection of anti-LASV glycoprotein (GP) human IgG and IgM antibodies specific to LASV lineage II, III and IV on stabilized pre-fusion GP antigens[43].

## PBMC isolation

Blood was collected by a trained phlebotomist at the KGH or in the survivor's town of origin. PBMCs were isolated from whole blood in the Lassa Laboratory at the Kenema Government Hospital (Kenema, Sierra Leone) or at the ISTH laboratory (Nigeria). Three volumes of PBS were combined with whole blood and layered on Ficoll-Paque (Fisher). The diluted blood was spun at $400 \times g$ (room temperature) without brake after which the mononuclear cell layer was isolated and washed twice with PBS. PBMCs were slowly frozen in a -80˚C freezer in RPMI 1640 medium (Gibco) containing 10% DMSO and 20% FCS. Frozen PBMCs were shipped to the United States in dry ice or a liquid nitrogen dry shipper and stored in liquid nitrogen until use.

## rscVSV preparation

Recombinant single cycle (rsc) VSVs encoding Lassa virus Josiah strain (Lineage IV) full length proteins (NP, GPC1, and GPC2) and their fragments (47–71 amino acids) were prepared by the method described previously by our laboratory[27, 32]. Briefly, viral DNA (see S1 Table for amino acid positions for each inserted sequence) was amplified in a polymerase chain reaction with gene specific primers and inserted into the pVSV-G-FLAG plasmid. LASV genes without stop codons are inserted upstream of the FLAG epitope which has its own stop codon. For LASV fragment sequences, naturally occurring methionine codons were used as start codons when possible (see S1 Table, blue font). Otherwise, the ATG start codon was added to the naturally occurring sequence. rscVSVs were produced and purified as previously described[27, 32].

## RT-PCR

BHK-21 cells (C-13; obtained from ATCC CCL-10,) were infected with rscVSVs encoding LASV proteins. RNA was isolated from cells after 6 hours of infection as previously described using TRI reagent and BCP phase separation techniques (Molecular Reseach Center, Inc)[32]. Oligonucleotide dT and SuperScript IV reverse transcriptase (Invitrogen) were used to make cDNA from isolated RNA. cDNA was amplified by PCR using Lassa gene (for forward primers) and FLAG epitope (for the reverse primer) specific oligonucleotides (listed in S3 Table) using GoTaq (Fisher) and separated by agarose gel electrophoresis.

## Western blotting

rscVSV infected BHK-21 cells were assessed for LASV protein expression at eight hours post-infection. Cell lysates were prepared as described previously[32] an separated on a 4–20% SDS-PAGE gel (Bio-Rad laboratories). Proteins were transferred to PDVF membrane (Millipore), blocked for 30 min at room temperature with TBS containing 0.05% Tween-20 (TBS-T) containing 5% skim milk, and then incubated with anti-flag rabbit polyclonal antibody (1:1,000) (Cayman Chemical Company). Horse radish peroxidase-conjugated anti-rabbit secondary antibody (1:1,000) (Pierce) was used with SuperSignal West Pico Chemiluminescent Substrate (Therm) and visualized by LAS-4000 system (GE Healthcare Life Sciences).

## T cell assay

PBMCs were infected with rscVSVs encoding full length or fragments of LASV proteins and EGFP at multiplicity of infection (MOI) of 15. To ensure T cell responsiveness in PBMC cultures, anti-human CD3 (OKT-3) (60 μg/ml) and CD28 (9.3) (20 μg/ml) antibodies were used as a positive control. After 4 hours, brefeldin A was added (4 μg/ml), and infected PBMCs were incubated overnight (total of approximately 16 hours) at 37˚C in 5% $CO_2$ PBMCs were washed in PBS and stained with anti-human brilliant violet 421 CD4 (RPA-T4) (BioLegend) and FITC CD8a (HIT8a) (Biolegend) for 1 h at 4˚C in FACS buffer (PBS containing 2% FCS and 0.2% Azide). Cells were washed in FACS buffer, fixed and permeabilized with BD Cytofix/Cytoperm (BD Biosciences) according to manufacturer's instructions. Intracellular staining with anti-human PE TNF-α (BD Biosciences), PE/Cy7 IFN-γ (4S.B3) (BD Biosciences), and APC IL-2 (MQ1-17H12) (BD Biosciences) antibodies followed for 1 h at 4˚C. Cells were washed and resuspended in FACS buffer for analysis using a LSR II (Becton Dickinson), and analyzed with FlowJo software (TreeStar, Inc). T cells incubated with peptides instead of rscVSVs were treated similarly except brefeldin A was added after one hour of peptide incubation (10 μg/ml; unpurified, Anaspec) and cells were stained four hours later.

## HLA typing

Genomic DNA was isolated using the Quick DNA miniprep Plus Kit (Zymo Research). HLA typing was performed as previously described[32]. Briefly, we used the TruSight HLA v2 Sequencing Panel (CareDx) according to manufacturer's protocols. The Illumina MiSeq platform was used to sequence final barcoded libraries. Results were analyzed using TruSight HLA Assign software (v2.1 RUO) and compared with sequences stored in the International ImMunoGeneTics Information System/HLA database (v3.37) to call HLA genotypes.

## MHC class I binding prediction

MHC-I Binding Prediction tool ((v2013-02-22) at the IEDB website (www.iedb.org) was used to determine putative T cell epitopes from deduced epitope sequences and HLA profiles (S2 Table). The recommended prediction settings (Consensus and NetMHCpan) were used and all 10aa peptides below the 2% prediction ranking were tested.

## Statistics

All statistics were calculated using GraphPad Prism and Microsoft Excel software. Tests performed are indicated in the figure legends. P values <0.05 were considered significant.

## Supporting information

**S1 Fig.** A) PBMCs from two control donors were infected with rscVSVs encoding for the indicated genes. At 6h post-infection, total RNA was isolated and cDNAs made from mRNA using oligo dT primers. Gene specific primers were used to amplify and assess expression of each gene. B) PBMCs from two control donors were infected with rscVSV encoding for EGPF or mock infected. After 6h post-infection, EGPF expression was assessed by flow cytometry in total PBMCs, T and B cells, granulocytes, and monocytes and compared to mock infected PBMCs.
(TIF)

**S2 Fig. Gating strategy for cells analyzed in this manuscript.**
(TIF)

**S3 Fig. CD8+ T cells from LF survivors were assessed for single, double, and triple cytokine expression after overnight stimulation with rscVSVs expressing EGFP, NP, GP1, GP1, and SSP-GP2.** Positive gates were defined as 1.2 log$_{10}$ fluorescence over the median negative control fluorescence as depicted in Fig 4A and 4B.
(TIF)

**S4 Fig. CD4+ T cells from LF survivors were assessed for single, double, and triple cytokine expression after overnight stimulation with rscVSVs expressing EGFP, NP, GP1, GP1, and SSP-GP2.** Positive gates were identical to those used for CD8+ T cells.
(TIF)

**S5 Fig. Comparisons of HLAs expressed by the Sierra Leonean and Nigerian Lassa fever survivors.**
(TIF)

**S6 Fig. Quantification of LASV-specific IgG in 29 Sierra Leonean LF Survivors.** Dotted line represents negative control value. Optical density values for an additional seven patients were obtained but could not be converted into U/ml. However, six of seven were considered positive based on negative control values.
(TIF)

**S1 Table. Amino acid positions of antigens encoded by rscVSVs.** Lines in blue indicate a start codon (ATG) was added to the sequence.
(PDF)

**S2 Table. Predicted peptides tested in the region of the deduced epitope and associated HLA profiles.**
(PDF)

**S3 Table. Primers used to identify expression from rscVSV infected cells.**
(PDF)

## Acknowledgments

We would like to thank the clinical staff at Kenema Government Hospital and the Lassa Ward, especially Francis Baimba for his expertise in phlebotomy as well as Simbirie Jalloh for managing the Lassa fever program at the KGH. This is manuscript #29897 from The Scripps Research Institute.

## Author Contributions

**Conceptualization:** Brian M. Sullivan, Saori Sakabe, Juan Carlos de la Torre, Pardis C. Sabeti, Kristian G. Andersen, Robert F. Garry, Donald S. Grant, John S. Schieffelin, Michael B. A. Oldstone.

**Data curation:** Brian M. Sullivan, Saori Sakabe, Karthik Gangavarapu, John S. Schieffelin.

**Formal analysis:** Brian M. Sullivan, Jessica N. Hartnett, Nhi Ngo, Mambu Momoh, Beatrice Cubitt, Refugio Robles-Sikisaka, Luis Branco.

**Funding acquisition:** John S. Schieffelin, Michael B. A. Oldstone.

**Investigation:** Brian M. Sullivan, Saori Sakabe, Jessica N. Hartnett, Nhi Ngo, Augustine Goba, Mambu Momoh, John Demby Sandi, Lansana Kanneh, Beatrice Cubitt, Selma D. Garcia, Brian C. Ware, Dylan Kotliar, Refugio Robles-Sikisaka, Karthik Gangavarapu, Philomena

Eromon, Ikponmwosa Odia, Ephraim Ogbaini-Emovon, Onikepe Folarin, Sylvanus Okogbenin, Peter O. Okokhere.

**Methodology:** Brian M. Sullivan, Saori Sakabe, Jessica N. Hartnett, Nhi Ngo, John Demby Sandi, Beatrice Cubitt, Dylan Kotliar, Refugio Robles-Sikisaka, Karthik Gangavarapu, Luis Branco, Juan Carlos de la Torre, Kristian G. Andersen, John S. Schieffelin.

**Project administration:** Brian M. Sullivan, Jessica N. Hartnett, Augustine Goba, Peter O. Okokhere, Christian Happi, Juan Carlos de la Torre, Pardis C. Sabeti, Kristian G. Andersen, Robert F. Garry, Donald S. Grant, John S. Schieffelin, Michael B. A. Oldstone.

**Resources:** Christian Happi, Robert F. Garry, Donald S. Grant.

**Supervision:** Brian M. Sullivan, Augustine Goba, Lansana Kanneh, Beatrice Cubitt, Philomena Eromon, Peter O. Okokhere, Christian Happi, Juan Carlos de la Torre, Pardis C. Sabeti, Kristian G. Andersen, Robert F. Garry, Donald S. Grant, John S. Schieffelin, Michael B. A. Oldstone.

**Validation:** Brian M. Sullivan, Saori Sakabe, Luis Branco.

**Writing – original draft:** Brian M. Sullivan, Saori Sakabe.

**Writing – review & editing:** Juan Carlos de la Torre, Pardis C. Sabeti, John S. Schieffelin, Michael B. A. Oldstone.

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
