## [Decision Letter · Decision Letter 0]

20 Nov 2019

Dear Dr. Sullivan,

Thank you very much for submitting your manuscript "High Crossreactivity of Human T Cell Responses Between Lassa Virus Lineages" (PPATHOGENS-D-19-01877) for review by PLOS Pathogens. Your manuscript was fully evaluated at the editorial level and by independent peer reviewers. The reviewers appreciated the attention to an important problem, but raised some substantial concerns about the manuscript as it currently stands. These issues must be addressed before we would be willing to consider a revised version of your study. We cannot, of course, promise publication at that time.

We therefore ask you to modify the manuscript according to the review recommendations before we can consider your manuscript for acceptance. Your revisions should address the specific points made by each reviewer.

(1) A letter containing a detailed list of your responses to the review comments and a description of the changes you have made in the manuscript. Please note while forming your response, if your article is accepted, you may have the opportunity to make the peer review history publicly available. The record will include editor decision letters (with reviews) and your responses to reviewer comments. If eligible, we will contact you to opt in or out.

(2) Two versions of the manuscript: one with either highlights or tracked changes denoting where the text has been changed; the other a clean version (uploaded as the manuscript file).

Additionally, to enhance the reproducibility of your results, PLOS recommends that you deposit your laboratory protocols in protocols.io, where a protocol can be assigned its own identifier (DOI) such that it can be cited independently in the future. For instructions see http://journals.plos.org/plospathogens/s/submission-guidelines#loc-materials-and-methods

We hope to receive your revised manuscript within 60 days. If you anticipate any delay in its return, we ask that you let us know the expected resubmission date by replying to this email. Revised manuscripts received beyond 60 days may require evaluation and peer review similar to that applied to newly submitted manuscripts.

[LINK]

Sincerely,

Alexander Bukreyev, Ph.D.

Associate Editor

PLOS Pathogens

Susan Ross

Section Editor

PLOS Pathogens

Kasturi Haldar

Editor-in-Chief

PLOS Pathogens

orcid.org/0000-0001-5065-158X

Grant McFadden

Editor-in-Chief

PLOS Pathogens

orcid.org/0000-0002-2556-3526

Reviewer's Responses to Questions

**Part I - Summary**

Reviewer #1: Sullivan et. al., are seeking to determine the nature of T-cell response in survivors of Lassa fever. To do this they used recombinant single cycle Vesicular Stomatitis Virus (rscVSV)-G pseudotyped viruses, encoding complete or partial fragments of the Lassa virus (LASV) (lineage IV) nucleoprotein (NP) and glycoprotein precursor protein (GPC), to infect PMBCs from Lassa fever survivors from both Nigeria and Sierra Leone (LASV, presumably lineages II and IV). They then measured CD4+ and CD8+ T cell responses in the VSV-infected PBMCs of the survivors and found that all survivors were able to mount T-cell responses towards lineage IV proteins, regardless of their country of origin. This suggests that lineages may share T-cell epitopes that might be exploited for vaccine targeting. However, there are significant reservations relating to the underlying method being applied to measure T cell responses, the absences of key pieces of data from figures, and the lack of confirmatory analysis using peptide-based assays. The net result is that it is difficult to determine the validity and impact of the results without more experimental data being provided.

Reviewer #2: Sullivan et al provide a collection of data supporting the cross clade responses of T-cells as observed from LASV survivors from both Nigeria and Sierra Leone. The study has immediate importance as the sample populations are from humans and thus has immediate relevance with respect to interpretation; however, is not without caveats associated with this kind of study (ie unknown time since or severity of infection, length of convolescence, ect...) of which the authors adequately recognize. The major findings suggest conserved T-cell epitopes common across phylogeographic regions and isolates, a finding rightly recognized as novel. While there are also caveats in the methodology utilized, I believe the authors adequately addressed the shortcomings (i.e. use of recombinant VSV versus other methods). This work will be of great interest to those studying arenavirus immunology, specifically those interested in old world arenavirus infections as well as those interested in developing vaccines against these high priority agents.

Reviewer #3: Lassa virus is the causative agent of Lassa Hemorrhagic Fever (LF), with high morbidity and mortality in West Africa.  Currently, four lineages of the Lassa virus circulate in West Africa, although only lineage IV is the main target in the most current vaccine candidates. Therefore, It is important to assess if the Lassa-specific immune response is able to cross-recognize antigens deriving from the four different lineages. Following this concept, in this manuscript, Sullivan et al. sought to determine the memory T cell reactivity against Lassa lineage IV antigens in Lassa infected survivor from Nigeria (where only lineages I-III circulate) and Sierra Leone (where only lineage IV circulate).  The authors found no difference in terms of T cell reactivity between Nigerian and Sierra Leone infected survivors, suggesting that Nigerian memory T cells are able to cross-recognize lineage IV with a magnitude of response comparable to the ones observed in Sierra Leone survivors where only lineage IV circulation is observed. These results have important implications in Lassa vaccination design however, a more in-depth characterization of T cell reactivity, as well as additional information on the specific epitopes, is required. 

**Part II – Major Issues: Key Experiments Required for Acceptance**

Reviewer #1: Specific concerns:

1. To determine T cell responses to Lassa antigens, the experiments presented utilize recombinant single cycle Vesicular Stomatitis Virus (VSV) pseudotyped with its native glycoprotein, G, added in trans. The genomes of these viruses have had the gene for G replaced with the Lassa NP, NP fragments, GPC, GPC fragments or GFP. In the first experiments, viruses encoding larger gene fragments were individually incubated with PBMCs from survivors for four hours, followed by the addition of brefeldin A (~16 hours) and then the cells were fixed, permeabilized and CD4+ or CD8+ T cell intracellular cytokine staining for TNF-a and IFN-g was measured using flow cytometry. Subsequent experiments used smaller overlapping gene fragments to “map” the T cell epitopes. There are several concerns about the use of this method and the control data that are presented:

a. The assay system uses bulk PBMCs infected by the recombinant single cycle VSV (rscVSV) viruses to present Lassa antigens via MHC. However, there is no evaluation of the ability of rscVSV to transduce PBMCs or the types of cells that are being transduced. Instead transduction, transcription and translation of Lassa proteins is shown in the BHK-21 kidney cell line (see Figure 1B). This cell line is highly permissive to VSV infection and are often used for the generation of pseudotyped VSV virus. Since efficient transduction of antigen presenting cells is a key component of the assay, they should show transduction, transcription and protein expression in human PBMCs samples. Ideally this would be done for each survivor, but given the likely limited amount of PBMCs available, it is reasonable to perform this analysis on several naïve human samples to show consistency of transduction, transcription and expression levels between subjects.

b. In each of the figures with T-cell responses, there is no representation of the “no stimulation” and the negative control (GFP virus) on the charts. Similarly, there are no representations of the flow cytometry images for the figures. These data are critical to the interpretation of the data and to determine the validity of the analysis. These controls and panels were provided in the authors previous PNAS manuscript on Ebola (PNAS. 2018. 115(32:e7578-86) and should be provided in this manuscript as well.

c. Similarly, since the results and the analysis are looking at very low frequency events, gating strategies can have a significant impact on the results. In addition to the information requested in 1b (above), please provide a supplemental figure showing representative gating trees for each of the unique strategies used.

d. The multiplicity of infection (moi) of 15 is very large and means that >99.5% of the PBMCs would be infected more than one time, including T cells (J Exp Med. 1978. 148(4):837-49). Given the critical dependence of the T-cell assays on both T-cells and APCs, it is important to demonstrate which cell types are being infected and to what extent. This analysis could be performed on several naïve human samples using rscVSV-GFP pseudotype viruses.

e. Despite there being staining for TNF-a, IFN-g and IL-2 by intracellular staining, data is only shown for magnitude of TNF-a/IFN-g double positive cells. Data on the magnitude and proportion of single, dual and triple positive intracellular cytokine responses should be discussed and presented, even if they were not significantly different than the GFP control virus.

2. Putative T-cell epitopes are identified as candidates and there is no confirmation using peptide-based assays. This is an important confirmation that should be performed, and the data provided in a main figure. Also, there is a lack of a discussion/figures showing the locations of the identified putative epitopes, their conservation between lineages and what the overlap or lack of overlap means in the context of next steps for a vaccine. It would helpful to provide this information in the text and a figure to accompany the analysis that shows the sequences from the lineages I-III and their alignment with the lineage IV sequence identified in this study.

Reviewer #2: None, the study was well done.

Reviewer #3: T cell reactivity is determined analyzing only polyfunctional T cells IFNγ+TNFα+ double producers, however, it is not clear if these polyfunctional T cells represent the majority of T cell responses, or if the instead stronger magnitude of responses is observed when looking at single IFNγ or TNFα producer.

Does the frequency of responders increase if the single producers are considered?

Additionally, in the methods, authors also stain for IL-2 production but do not show any data relating to this issue and do not report whether they the observe triple producer T cells.

Authors dissect T cell reactivity by using fragments spanning 60aa length, and they deduce CD8 epitopes by identifying as epitopes the overlapping region. This approach is not suitable to define the CD8-T cell epitopes with sufficient accuracy. I would suggest either removing this analysis or , based on mapping the potential epitopes at least in silico by the combined use of HLA typing information and bioinformatic predictions. 

**Part III – Minor Issues: Editorial and Data Presentation Modifications**

Reviewer #1: Other concerns to address:

3. Many in the field are using overlapping bulk peptides and peptide pools to map CD4+ and CD8+ T-cell epitopes. Despite being more costly, use of peptides avoids the confounding effects of using viruses that are capable of inducing cytokine responses and infecting all the cells being used in the assay, including T cells. Please provide a few sentences on why the rscVSV approach was chosen, it’s limitations and benefits compared to the peptide method.

4. Abstract: Line 45-46 mentions four viral lineages with substantial inter- and intra-lineage diversity. However, there is substantial inter-lineage diversity, but the intra-lineage diversity isn’t considered to be substantial (Ehichioya DU et al. Phyelography of Lassa virus in Nigeria. J Virol. 2019.93(21)). In fact, the idea that underlies the notion of having a single vaccine that is protective against all the lineages rests on the inter-lineage diversity.

5. On page 6 (lines 109 and 100) the number of subjects tested are 11 Nigerians and 37 Sierra Leoneans (total of 48). On lines 160 through 162, the authors describe the responses in these 48 individuals as being made up of 16 with both CD4 and CD8 responses, 13 with just CD4 responses and 20 with just CD8 responses (total = 49). Please correct the numbers.

6. Introduction: Line 86 – 87. Recommend changing “…the western edge of continental Africa to Nigeria” to “West Africa” as what is being described is more properly referred to as West Africa.

7. Results: Line 126 LCMV should be changed to LASV.

8. Line 137 Fig 1A & 1C. There was no label for lane 4.

9. Line 171. “…LASV strains from lineages II and III.” The manuscript assumes the strains of the survivors were likely lineages II and III. However, current literature on the molecular epidemiology of LASV shows that LASV strains in Irrua where the Nigerian survivors were from, are lineage II strains [Ehichioya DU et al. Phyelography of Lassa virus in Nigeria. J Virol. 2019 Oct 15;93(21)]. Thus, in the absence of confirmed LASV sequences matched to each survivor, it is best to assume the survivors were infected by lineage II viruses.

10. Line 163, 165, 168 and Figure 2B. In these lines and figure panel, the manuscript shows a combined NP & GPC. It is not clear whether this is (1) the result of two separate PBMC infections added together OR (2) if PBMCs were infected simultaneously with rscVSV NP and rscVSV GPC

11. Materials & methods:

a. Residue numbers, cloning or synthesis method and other details specific to the vector production should be provided in the section on the rscVSV preparation (line 330)

b. Line 337. Ebola should be changed to LASV.

c. Line 344. The forward primer sequences used for the PCR should be provided and the expected band size of the amplicon stated.

12. Figure 1A. It is unclear why there is a schematic for the entire L and S segments, would consider removing.

13. S.P should be S.S.P (stable signal peptide).

14. Figure 1B & C. While size markers are shown, there are no values given for the markers. Please label 3-4 of the major bands of the DNA and protein ladders, particularly those near bands of interest.

Reviewer #2: Lines 100-102: Consider revising or removing this sentence as it is not in the same context as natural infection. Vaccinia delivery of antigen is an artificial system that is not the same context as natural arenavirus infection. The importance of cell mediated immunity can be better argued another way, even using LCMV as a surrogate would be more appropriate. VSV delivery of LASV antigens induces a neutralizing antibody response (https://journals.plos.org/plosmedicine/article?id=10.1371/journal.pmed.0020183#s3), but at the same time does not imply the importance of humoral immunity in clearance of natural LASV infection, only that protection is possible via the context of VSV infection where LASV GPC is the surrogate glycoprotein. Please correct this.

The authors suggest a conserved region(s) of LASV antigens that both survivors from Sierra Leone and Nigeria recognize. Readers might benefit from knowing what level of phylogenetic conservation at nucleotide or amino acid level exist across clades in these "hot spot regions". Please consider including this to support your conclusions.

Reviewer #3: Figure 3C is not discussed in the results section.

PLOS authors have the option to publish the peer review history of their article (what does this mean?). If published, this will include your full peer review and any attached files.

Reviewer #1: No

Reviewer #2: No

Reviewer #3: No

---

## [Decision Letter · Decision Letter 1]

24 Jan 2020

Dear Dr. Sullivan,

We are pleased to inform you that your manuscript 'High Crossreactivity of Human T Cell Responses Between Lassa Virus Lineages' has been provisionally accepted for publication in PLOS Pathogens.

Before your manuscript can be formally accepted you will need to complete some formatting changes, which you will receive in a follow up email. A member of our team will be in touch within two working days with a set of requests.

Best regards,

Alexander Bukreyev, Ph.D.

Associate Editor

PLOS Pathogens

Susan Ross

Section Editor

PLOS Pathogens

Kasturi Haldar

Editor-in-Chief

PLOS Pathogens

orcid.org/0000-0001-5065-158X

Michael Malim

Editor-in-Chief

PLOS Pathogens

orcid.org/0000-0002-7699-2064

Reviewer Comments (if any, and for reference):

Reviewer's Responses to Questions

**Part I - Summary**

Reviewer #1: The authors have adequately addressed my concerns

Reviewer #2: The authors have sufficiently addressed all the concerns raised by this reviewer. I am also impressed by the efforts the authors made to address other reviewers.

Reviewer #3: (No Response)

**Part II – Major Issues: Key Experiments Required for Acceptance**

Reviewer #1: none

Reviewer #2: All addressed

Reviewer #3: See below

**Part III – Minor Issues: Editorial and Data Presentation Modifications**

Reviewer #1: none

Reviewer #2: All addressed

Reviewer #3: See below

PLOS authors have the option to publish the peer review history of their article (what does this mean?). If published, this will include your full peer review and any attached files.

Reviewer #1: No

Reviewer #2: No

Reviewer #3: No

---

## [Editor Report · Acceptance letter]

28 Feb 2020

Dear Dr. Sullivan,

We are delighted to inform you that your manuscript, "High Crossreactivity of Human T Cell Responses Between Lassa Virus Lineages," has been formally accepted for publication in PLOS Pathogens.

Best regards,

Kasturi Haldar

Editor-in-Chief

PLOS Pathogens

orcid.org/0000-0001-5065-158X

Michael Malim

Editor-in-Chief

PLOS Pathogens

orcid.org/0000-0002-7699-2064